# Clinical Islet Xenotransplantation: Development of Isolation Protocol, Anti-Rejection Strategies, and Clinical Outcomes

**DOI:** 10.3390/cells13100828

**Published:** 2024-05-13

**Authors:** Shinichi Matsumoto, Kyohei Matsumoto

**Affiliations:** 1Graduate School of Medicine, Kobe University, Kobe 650-0017, Japan; 2Medical Porcine Development Organization, Kobe 650-0017, Japan; 3Second Department of Surgery, Wakayama Medical University, Wakayama 641-0012, Japan; kyohei-m@wakayama-med.ac.jp

**Keywords:** islet xenotransplantation, allogeneic islet transplantation, type 1 diabetes, donor shortage, clinical trials

## Abstract

Allogeneic islet transplantation has become a standard therapy for unstable type 1 diabetes. However, considering the large number of type 1 diabetic patients, the shortage of donors is a serious issue. To address this issue, clinical islet xenotransplantation is conducted. The first clinical islet xenotransplantation was performed by a Swedish team using fetal pancreatic tissue. Thereafter, clinical trials of islet xenotransplantation were conducted in New Zealand, Russia, Mexico, Argentina, and China using neonatal pig islets. In clinical trials, fetal or neonatal pancreata are used because of the established reliable islet isolation methods. These trials demonstrate the method’s safety and efficacy. Currently, the limited number of source animal facilities is a problem in terms of promoting islet xenotransplantation. This limitation is due to the high cost of source animal facilities and the uncertain future of xenotransplantation. In the United States, the first xenogeneic heart transplantation has been performed, which could promote xenotransplantation. In Japan, to enhance xenotransplantation, the ‘Medical Porcine Development Association’ has been established. We hope that xenogeneic transplantation will become a clinical reality, serving to address the shortage of donors.

## 1. Introduction

Type 1 diabetes is a chronic metabolic disorder characterized by the destruction of insulin-secreting beta cells in the pancreatic islets [1]. The Diabetes Control and Complications Trial (DCCT) demonstrated that intensive insulin therapy could reduce HbA1c levels and dramatically reduce the progression of the microvascular complications of diabetes, including nephropathy, neuropathy, and retinopathy, when compared to conventional insulin therapy [2]. However, intensive insulin therapy increased the number of hypoglycemia events, including life-threatening, severe hypoglycemic events [3]. In fact, severe hypoglycemic events cause substantial morbidity and mortality in type 1 diabetic patients, while allogeneic islet transplantation substantially reduces the number of severe hypoglycemic events, with stable HbA1c levels [4].

A successful clinical allogeneic islet transplantation was described in 2000 by the University of Alberta group (Edmonton protocol) [5], and the Alberta Government rapidly promoted clinical islet transplantation, which became a government-funded, non-research ‘standard-of-care therapy’ on April 1st 2001 [6]. Since then, allogeneic islet transplantation has been pioneered by the Canadian group. A successful phase 3 clinical islet transplantation was described by Hering and colleagues in 2016 [4]. This islet transplantation was intended to prevent severe hypoglycemia with stable blood glucose controls [4]. In the US, in 2023, the Food and Drug Administration (FDA) approved allogeneic islets as a biological product [7]. Thus, allogeneic islet transplantation has become a standard therapy for unstable type 1 diabetic patients worldwide. However, considering the large number of type 1 diabetic patients, the shortage of human organ donors is a serious issue.

In order to address the problem of donor shortages, islet xenotransplantation using porcine islets has been performed clinically. In addition to addressing donor shortages, islet xenotransplantation has several advantages. Islet xenotransplantation can use healthy normal donor pigs, which can be comprehensively assessed regarding the status of pathogens. Furthermore, islet isolation and transplantation can be conducted on a routine basis. However, islet xenotransplantation also has disadvantages, including limited clinical experiences, possible zoonosis, the necessity for donor source facilities, and the need for appropriate ethical and regulatory frameworks (Table 1).

One of the unique advantages of xenogeneic islet transplantation is the stable islet isolation method, which shows no failure when using fetal or neonatal pancreata. In fact, the stable islet isolation method was one of the reasons for the performance of the first islet xenotransplantation using fetal pancreata [8]. As a matter of fact, even with the advanced human islet isolation method, the success rate of human islet isolation was found to be only 52.5% (170/324) [9]. Therefore, understanding the development of the fetal or neonatal porcine islet isolation method is important.

On the other hand, overcoming xenogeneic rejection is challenging. Anti-rejection drugs, encapsulation, and Sertoli cells were used for clinical islet xenotransplantation with modest outcomes.

In this review article, at first, we focus on the development of the porcine islet isolation protocol. Thereafter, anti-rejection strategies and clinical outcomes are discussed. Lastly, we mention ethical and regulatory considerations related to clinical xenotransplantation.

## 2. Development of Islet Isolation Protocols

### 2.1. Development of the First Clinical Porcine Islet Isolaition Protocol

The first clinical trial of porcine islet transplantation was conducted by a Swedish team in the 1990s [10]. Isolated islets from porcine fetal pancreata were identified as islet-like cell clusters (ICCs). In fact, in the 1970s, it was demonstrated that fetal pancreas tissue transplantation reversed diabetes in rodent models [11,12]. Based on rodent models with fetal pancreas tissue transplantation, Sandler and colleagues developed a method to obtain human islet-like cell clusters (ICCs) from the human fetal pancreas ([13], Table 2). Finally, Korsgren and his colleagues applied this method to fetal pig islet isolation.

### 2.2. Human Islet-like Cell Cluster (ICC) Protocol

The basic idea of creating ICCs is to procure a pancreas from a fetal donor, mince the donor pancreas into pieces, digest the minced pancreas, and culture the digested pancreatic tissue [13]. After culturing the digested tissue, exocrine tissue typically diminishes, and only endocrine tissue becomes ICCs.

The Swedish group demonstrated their protocol to generate ICCs with pancreata from 10–26-week-old fetuses after abortion [13]. The procured pancreata were minced in sterile Hanks’ solution (HBSS) into fragments of approximately 3–4 mm^3^. The minced pancreatic tissue was suspended in RPMI medium 1640 and kept at 4 °C during transport to the laboratory. In the laboratory, the minced pancreas was transferred to a glass vial containing 5 mg of collagenase dissolved in 2 mL of HBSS and supplemented with antibiotics. To digest the minced pancreata, the glass vial was shaken in a 37 °C water bath for 5–10 min. The digestion was halted when the tissue fragments were almost completely disintegrated and barely visible under visual inspection. The digested tissue was transferred to plastic tubes and washed twice with HBSS. The pelleted digest was resuspended in a culture medium composed of PRMI 1640 with antibiotics plus heat-inactivated fetal calf serum (FCS) for culture. The culture dishes were kept at 37 °C in an atmosphere of 5% CO_2_ in humidified air. The culture medium was changed after 48 h and then every day for the next 7 days. After the completion of this process, the team successfully obtained human-beta-cell-containing cell aggregates. They named the outgrowing beta-cell-containing cell aggregates ‘islet-like cell clusters (ICCs)’.

### 2.3. Korsgren Porcine Islet-like Cell Cluster Protocol

Korsgren and colleagues applied the method of human ICC to fetal pig pancreata ([14], Table 2). Pregnant sows were used to obtain fetuses that were 51 to 77 days old. The procured pancreata were placed on chilled HBSS supplemented with antibiotics within 1 h of killing the sow. Each of the pancreata samples used (2–4/L) was minced into 1–2 mm^3^ pieces and transferred to a sterilized glass vial containing 100 mg of collagenase dissolved in 8 mL of HBSS. After vigorous shaking in a 37 °C water bath for 5–6 min, when the tissue fragments had nearly disintegrated, digestion was arrested by adding 10 mL of HBSS. After washing the digest twice in HBSS, it was then resuspended in RMPI 1640 culture medium. Tissue culture dishes were used to distribute the aliquots in the RPMI 1640 culture medium. Finally, fetal calf serum (FCS) or heat-inactivated pooled human serum (HS) was added. The culture dishes were maintained at 37 °C in a gas phase of 5% CO_2_ in humidified air. The culture medium was changed every two days until the sixth or seventh day. The investigation revealed that HS had the ability to increase the number of ICCs in comparison to FCS. A total of more than 100,000 ICCs were produced from 3 litters, at an age of 67–77 days, after a culture in the presence of HS. The functional capacity of the ICCs was not yet mature. However, the researchers implanted the ICCs under the kidney capsules of non-diabetic nude mice, and, after 4 weeks, they found frequent insulin- and glucagon-positive cells.

Elliott and colleagues modified this method for their first clinical islet xenotransplantation [15]. They used the same digestion method, but they introduced nicotinamide for the culture process. This method was used for the first New Zealand clinical trial.

### 2.4. Korbutt Protocol for Neonatal Porcine Islet Isolation

In 1996, Korbutt and colleagues at the University of Alberta published a method for the large-scale production of porcine neonatal islet cells ([17], Table 2). They used 1- to 3-day-old Landrace–Yorkshire neonatal pigs (1.5–2.0 kg body weight). The pancreas was retrieved and preserved in cold HBSS supplemented with 0.25% BSA, 10 mmol/L of HEPES, 100 U/mL of penicillin, and 0.1 mg/mL of streptomycin. The warm ischemic time was less than 10 min. Each of the pancreas samples was cut into fragments of approximately 1–2 mm^3^, transferred to sterile tubes containing HBSS supplemented with 2.5 mg/mL collagenase, and gently agitated for 16–18 min in a shaking water bath at 37 °C. The digested tissue was filtered through a 500 μm nylon screen, washed four times in HBSS, and placed in Petri dishes containing Ham’s F 10 tissue culture medium (10 mmol/L of glucose, 50 μmol/L of IBMX, 0.5% BSA, 2 mmol/L of L-glutamine, 10 mmol/L of nicotinamide, 100 U/mL of penicillin, and 100 μg/mL of streptomycin). Culture dishes were maintained at 37 °C (5% CO_2_, 95% air) in humidified air for 9 days, with the medium and dishes changed the first day after isolation. Subsequently, the medium was changed every two days. The researchers were able to obtain 50,000 islet cell aggregates, consisting of primarily epithelial cells and pancreatic endocrine cells.

### 2.5. Hillberg’s Modification of the Korbutt Method

Hillberg and colleagues in New Zealand modified the Korbutt method for porcine neonatal islet isolation ([18], Table 2). They used 7- to 10-day-old neonatal piglets as pancreas donors. The pancreata were cut into pieces, and the pieces were digested with 0.75 mg/mL of Liberase MTF C/T and thermolysin. A Ricordi chamber was introduced to facilitate digestion [19]. A 300 μm stainless-steel screen was used to filter the digested tissue. Free neonatal porcine islet-like clusters (NPICs) were cultured in 500 mL disposable spinner flasks at a concentration of 40 mL per gram of pancreatic tissue in RPMI-CAN serum-free medium (RPMI-1640, 0.3 g/L of ciprofloxacin, 1% human serum albumin, 0.12% Wt/Vol nicotinamide). In a 5% CO_2_ incubator at 37 °C with humidity, the flasks were stirred at 80–90 rpm. The culture medium was replaced daily (100%) using the 15 min settlement process in 50 mL tubes. They assessed viability using an AO stain. The viability was 98%, 99%, and 99% after 1, 2, and 4 weeks in culture, respectively.

This method was used for the Russian clinical trial [20], the next series of New Zealand clinical trials [21,22], the Argentine clinical trials [22,23], and the Mexican clinical trial [24,25,26]. For the Argentine clinical trial, ETK solution was introduced for pancreas preservation, which resulted in improved quality and a greater quantity of isolated islets [22,23].

### 2.6. Valdes’s Modification of the Korbutt Method

Vales and his collages modified the Korbutt method as follows [27]. Male 7–10-day-old piglets were used for pancreatic and testicular procedures. The pancreata were minced and placed in a flask with three glass beads that contained 2 mL/g of digestion solution (Liberase HI 1.5 mg/mL, 150 μ/mL of DNase I, 27 μg/mL of xylocaine in HBSS). The flask was placed in an agitation incubator at 140 rpm for 15 min at 37 °C. The digested tissue was placed onto a 400 μm stainless-steel mesh. Undigested tissue was digested again with half of the digestion solution for 7 min. The digested tissue was washed at 200 g for 3 min, three times, with HBSS supplemented with 0.2% human serum albumin. Then, the pellets were resuspended in RPMI 1640 medium supplemented with 2% human serum albumin and 10 mM of nicotinamide for culture at 37 °C with 5% CO_2_. The media were changed at 24 and 48 h.

### 2.7. Wei Islet Isolation Protocol

Between 2013 and 2017, the Wei Wang team conducted clinical islet xenotransplantation in 10 type 1 diabetic patients [28]. The third Xiangya Hospital (Changsha, China) and Hunan Xeno Biological Technology Co., Ltd. (Changsha, China) have established a Designated Pathogen-Free (DPF) donor facility to produce clinical-grade pigs. Islets were isolated from neonatal pig pancreata and cultured with their modified culture medium. The detailed protocol has not been published.

### 2.8. Discussion and Conclusions Regarding the Islet Isolation Method

The first clinical trial of the Swedish group used ICCs due to the stable islet isolation method. In general, the success rate of human islet isolation is approximately 50% [9], and adult porcine islet isolation is more difficult compared to human islet isolation [29]. Stable islet isolation is critically important for large-scale clinical trials or even for industrialization in the future. Therefore, understanding the development of fetal or neonatal porcine islet isolation is important.

In order to increase the islet yield, Korbutt and colleagues used a neonatal pancreas instead of a fetal pancreas. This is simply because the size of the neonatal pancreas is bigger than the fetal pancreas; however, this step has significant impact to provide enough islets with less donors. Importantly, the Korbutt islet isolation method has become a current gold standard for neonatal porcine islet isolation.

The New Zealand group initially used fetal pancreas and then introduced the Korbutt protocol to increase the islet yields and provide neonatal porcine islets for several clinical trials. Because the quality and quantity of islets are critically important, they continuously improve the neonatal porcine islet isolation method. Further modifications, including the use of ETK solution for pancreas preservation, the Ricordi method for pancreas digestion, and nicotinamide for culture, are important advancements in neonatal porcine islet isolation.

In conclusion, the current neonatal porcine islet isolation method could provide stable, high quantities of quality islets for clinical trials.

## 3. Anti-Rejection Strategies and Clinical Outcomes

### 3.1. Introduction

To prevent xenogeneic rejection, three strategies have been used, namely, anti-rejection drugs, encapsulation, and Sertoli cells. All strategies have shown positive outcomes; however, further improvement is desirable.

### 3.2. Anti-Rejection Drugs

#### 3.2.1. Introduction to Anti-Rejection Drugs

To prevent xenogeneic rejection, anti-rejection drugs were used by two teams. The first islet xenotransplantation was conducted on type 1 diabetic patients after allogenic kidney transplantation. Therefore, these patients had already received immunosuppressive drugs. To avoid xenogeneic rejection, the doses of the immunosuppressive drugs were increased. For the Chinese trial, they used autologous regulatory T cells, in addition to the regular immunosuppressive drugs. Both outcomes were promising.

#### 3.2.2. The First Swedish Cases

The patients undergoing the first islet xenotransplantation had undergone kidney transplantation; therefore, all patients received immunosuppressive drugs ([8,10], Table 3). Five patients received cyclosporine, prednisolone, and azathioprine; two patients received cyclosporine and prednisolone; and one patient received azathioprine and prednisolone. At the time of islet transplantation, the doses of the immunosuppressive drugs were temporarily increased. In addition, adjunctive therapy with 200 mg of anti-thymocyte globulin (ATG) was given daily for 1 week to the first five patients. The next three patients received 4 mg/kg of 15-deoxyspergualin (DSG) for 5 days. In the last two patients, the immunosuppression consisted of cyclosporine, azathioprine, prednisolone, and DSG.

For the first eight patients, the ICCs were injected intraportally, and, in the last two patients, the ICCs were placed under the kidney capsule of the renal graft. The first eight patients received 330,000–520,000 ICCs, and the last two patients received 800,000 and 1,000,020 ICCs.

Porcine C-peptide was secreted in the urine by four patients who received intraportal transplantation for 200–400 days (Table 2). In one renal-graft biopsy specimen, morphologically intact epithelial cells were stained positively for insulin and glucagon in the sub-capsular space. It was concluded that porcine pancreatic endocrine tissue can survive in the human body. However, no apparent clinical benefit was shown. Nevertheless, this group opened the door for islet xenotransplantation.

#### 3.2.3. Chinese Cases

Between 2013 and 2017, the Wei Wang team conducted clinical islet xenotransplantation in 10 type 1 diabetic patients ([28], Table 3). The third Xiangya Hospital and Hunan Xeno Biological Technology Co., Ltd. have established a Designated Pathogen-Free (DPF) donor facility to produce clinical-grade pigs. Islets were isolated from neonatal pig pancreata and cultured with their modified culture medium. They transplanted 10,000 IEQ/kg body weight islets into the patient’s liver.

They used immunosuppressive drugs and autologous regulatory T cells (Tregs) for immunosuppression. The patients received 0.087 mg/kg of Tacrolimus twice per day, mycophenolate mofetil 1 g twice per day, and belatacept prior to transplantation, on day 5, and at weeks 2, 4, 8, and 12, at a dose of 10 mg/kg. The dose of autologous Tregs was 2 × 10^6^ cells/kg.

It was stated that the patients’ conditions improved substantially, and the comprehensive score of transplantation was higher than that in a similar Japanese experiment [28]. This group demonstrated that porcine islets could survive with their immunosuppressive protocol and provide clinical benefits for type 1 diabetic patients.

### 3.3. Encapsulation Protocol

#### 3.3.1. Introduction

Based on the successful encapsulated human islet transplantation [30], the New Zealand group applied the encapsulation protocol for neonatal porcine islets [15,16]. The encapsulation protocol was originally established using a canine model.

#### 3.3.2. Soon-Shiong Method for Microencapsulation

Soon-Shiong and colleagues developed a method for microencapsulation using a canine model [31,32]. Purified canine islets were placed in a 1.6% solution of ultra-pure alginate with guluronic acid content ≥64%, and alginate droplets were generated by applying a coaxial air stream. Gel-entrapped islets were achieved via the ionic crosslinking of guluronic acid in a 0.4% W/V calcium chloride bath. The alginate capsules with positively charged polylysine (molecular weight 17,000) allowed for an efficient ionic interaction, followed by an outer coat of 0.1% alginate, providing a biocompatible capsule. They successfully treated seven (7/7) spontaneous diabetic dogs with the intraperitoneal transplantation of the encapsulated canine islets. This group applied this method for human islet transplantation and achieved insulin independence in type 1 diabetic patients [30]. The New Zealand group used their method for the first clinical islet xenotransplantation [15,16].

#### 3.3.3. Calafiore Microencapsulation Method

Calafiore and colleagues described an encapsulation method for improved biocompatibility using alginate and ornithine, instead of alginate and lysine [33]. Washed islets were mixed with 1.6% highly purified sodium alginate that was free of endotoxins and pyrogens. Droplets were generated via a combination of air shears and mechanical pressure using a peristaltic pump. The micro-droplets containing the islets were collected in a CaCl2 bath and transformed into gel microbeads. The beads, containing 1–2 islets and measuring an average of 500 µm in diameter, were sequentially double-coated with 0.12% and 0.06% poly-L-ornithine and, finally, with 0.04% sodium alginate. This encapsulation method was used for allogeneic islet transplantation without immunosuppression, and it was demonstrated to be clinically effective and safe [33,34].

The New Zealand group used this method for all clinical trials except for the first trial.

#### 3.3.4. Clinical Trials by the New Zealand Group

##### The First New Zealand Clinical Trial

In 1996, Elliott and colleagues transplanted encapsulated porcine islets into two type 1 diabetic patients ([15,16], Table 3). They applied the Korsgren protocol for porcine islet isolation ([17], Table 2). The isolated islets were encapsulated using the Soon-Shiong method [30,31,32].

In May 1996, a 41-year-old type 1 diabetic patient was enrolled in a xenotransplantation study approved by the Ministry of Health, New Zealand Government and by the Regional Ethics Committee [15,16]. The patient was free of diabetic complications; however, he had poor blood glucose control. The average daily insulin level was 53 units/day, and the HbA1c level was 9.3% in the 2 months prior to transplantation. In total, 15,000 IEQs/kg body weight of encapsulated porcine islets were placed in the peritoneal cavity via laparoscopy under general anesthesia. The average daily insulin dose was reduced by up to 30% for 14 months following the transplant. The HbA1c levels decreased to 7.1% in 2000 and 8.4% in 2005. Urinary porcine C-peptide was detected in the first month after transplantation, but it was undetectable after 14 months. Hypoglycemic episodes were reduced from 7–9 episodes per month to 0–4 episodes per month at 8 weeks after transplantation, and this effect was maintained for at least one year. According to the patient, in March 2005, his blood glucose control was still better than it was before transplantation. Intact encapsulated islets were discovered after performing biopsies.

The second patient was a 39-year-old female with a 24-year history of type 1 diabetes. She received maintenance immunosuppression (cyclosporine, azathioprine, and prednisone) for a successful renal transplantation that took place 2 years prior to this transplantation. The amount of islets that she received was 15,000 IEQ/kg. The daily insulin dose reduction was less than that of the first patient; however, her urinary porcine C-peptide levels were similar to those of the first patient.

Both patients experienced a significant reduction in the frequency of severe hypoglycemic reactions. Neither patient had any unusual symptoms or developed antibodies to the pig viruses tested.

In addition, four type 1 diabetic patients received non-encapsulated porcine islets in New Zealand, and no infections were found [15].

##### Russian Trial

In 2007, a pilot study with eight patients was approved by the Scientific and Ethics Committees of the Sklifosovsky Institute in Moscow, Russia [20]. The New Zealand team provided encapsulated islets. To confirm insulin deficiency in the patients before transplantation, the stimulated plasma C-peptide levels were confirmed to be less than 0.2 ng/mL. There were two groups consisting of 5000 IEQ/kg transplantation (*n* = 5) and 10,000 IEQ/kg (*n* = 3). Seven patients were followed for daily insulin dose and HbA1c up to 6 months (Figure 1 upper graph). Patients A, B, C, D, and E received 5000IEQ/kg, and F and G received 10,000 IEQ/kg.

Patients B achieved insulin independence at 3 months, and patient F achieved insulin independence at 6 months. Patient A reduced the insulin doses by more than 30%. The other patients had minimal changes in their daily insulin doses.

All patients except for patient D experienced a reduced HbA1c after transplantation (Figure 1 lower graph). Considering insulin dose and HbA1c reduction together, all patients except for patient D had a clinical benefit.

##### Series of New Zealand Clinical Trials

The clinical study was approved by the Minister of Health under the regulatory framework for xenotransplantation and the Northern Regional Ethics Committee. At Middlemore Hospital (Auckland, New Zealand), encapsulated neonatal porcine islets were given to fourteen type 1 diabetic patients (5000 IEQ/kg × 4 group 1, 10,000 IEQ/kg × 4 group 2, 15,000 IEQ/kg × 4 group 3, 20,000 IEQ/kg × 2 group 4) [19,21]. There were four serious adverse events, including hypersensitivity, post-procedural discomfort, anxiety, and a depressed mood. Although the treatment for depression was continued, all of these events were resolved without any residual effects. The number of unaware hypoglycemia events was reduced at all doses (Table 4). Average insulin doses were also reduced, except for group 4, and average HbA1c levels were also reduced, except for group 3 (Table 4). It was noted that one patient in group 1 reduced their insulin doses to less than half and maintained an HbA1c less than 7% [21].

##### Argentine Trial

At Hospital Eva Peron de San Martine (Buenos Aires, Argentina), clinical trials were conducted by the New Zealand group [23]. For this trial, they introduced an ETK solution for pancreas preservation before isolation, which improved the quality of the islets. Encapsulated islets were transplanted twice. The second transplantation was conducted 3 months after the first transplantation. Four type 1 diabetic patients received 5000 IEQ/kg encapsulated islets twice (group 1), while another four patients received 10,000 IEQ/kg encapsulated islets twice (group 2). When comparing values pre-transplant and 600 days after transplantation, the number of unaware hypoglycemic events was reduced in group 2 (Table 5). Average daily insulin doses and average HbA1c were reduced in both groups (Table 5).

In addition, 14 type 1 diabetic patients were given 10,000 IEQ/kg encapsulated islets twice [23]. The opinions of 21 Argentine patients 10 years after islet xenotransplantation were also recently published [35]. Because immunosuppressants were not used, cancer did not occur in any patient. Based on the patients’ opinions, compared with pre-transplantation, 15/21 (71%) patients reported improved diabetes management, 16/21 (76%) patients reported improved blood glucose levels, 18/21 (86%) patients reported a reduced number of severe hypoglycemia episodes, and 16/21 (76%) patients reported a reduced number of hyperglycemia episodes requiring hospitalization, even 10 years after transplantation. It seems that beta cell supplementation without immunosuppression could provide a long-term benefit for unstable type 1 diabetic patients.

### 3.4. Transplant Device with Sertoli Cells

#### 3.4.1. Introduction of Transplant Device with Sertoli Cells

The immunomodulating activity of testicular Sertoli cells for use in islet transplantation was first described by Selawry and colleagues in 1985 [36]. Sertoli cells secrete TGF-beta 1, which protects islet beta cells from autoimmune destruction and also converts islet-infiltrating cells from a beta-cell-destructive (IFN-gammna+) phenotype to a nondestructive (IL-4+) phenotype [37]. To conduct the co-transplantation of islets and Sertoli cells, the researchers developed a device. The device was composed of 6 × 0.8 cm surgical-grade stainless-steel mesh tubes and an interior rod constructed from polytetrafluoroethylene (PTFE) [38]. The devices were left in the abdominal region for two months to allow the formation of vascularized collagen tissue, which completely surrounded and penetrated the devices. Once the interior rod was removed, islets could be transplanted into the space surrounding the vascularized collagen tissue.

#### 3.4.2. Sertoli Cell Isolation Protocol

The method for neonatal porcine Sertoli cells (NPSC) was conducted as follows.

Under general anesthesia, the piglets were killed by exsanguination [38]. Then, the testis was aseptically removed and placed in cold washing solution (HBSS supplemented with 0.2% human serum albumin). Testis samples were minced and then washed three times with the washing solution. The washed testis samples were placed in a flask with 10 mL HBSS with 3.8 mg/mL Liberase HI, 150 μ/mL DNase I, and 27 μg/mL Xylocaine per gram of testis. The flask contained glass beads measuring 5 mm in diameter and was shaken at 140 oscillations per minute at 37 °C for 40 min. The tissue was sieved through a 400 μm stainless-steel mesh and washed twice at 150 g for 5 min. The washed pellet was resuspended in 1 M glycine and 2 mM EDTA HBSS, and washed again with DME at 150 g for 5 min. The final pellet was mixed with 1:1 (*v*/*v*) DME-F10 medium and 2% HAS and 1.22 g/L nicotinamide. Neonatal porcine Sertoli cells (NPSCs) were cultured at 140,000 cells/cm^2^ density at 37 °C with 5% CO_2_; then, after 24 h, the medium was changed.

#### 3.4.3. Clinical Trials of Neonatal Porcine Islets and Sertoli Cells in Subcutaneous Devices

In 2000, the first clinical trial in Mexico was conducted. This study included twelve adolescent type 1 diabetic patients. Two devices were inserted into the upper anterior wall of each patient’s abdomen subcutaneously. The devices were surrounded and penetrated by collagen tissue two months after they were placed. After removing the PTEF rod, the researchers implanted 250,000 islets and 30–100 Sertoli cells. From 6 to 9 months later, all patients except one received a second transplant into the previously implanted new device. Two patients achieved an insulin-free status and one patient remained insulin-free for an extended period [24].

The second clinical trial in Mexico was conducted in 2002–2003 [25]. Eleven adolescents with type 1 diabetes were treated using the same protocol as in the first trial. The HbA1c level, insulin dose, and diabetic chronic complications, including neuropathy, retinopathy, and nephropathy, were evaluated after 5 years [26]. Compared to the baseline conditions, the average values of HbA1c decreased significantly after the first and second transplantations, and the average doses of insulin decreased significantly after the first, second, and third transplantations. Nervous conduction was used for the neuropathy assessment, non-proliferative retinopathy was used for the retinopathy assessment, and micro-albuminuria was used for the nephropathy assessment. Neuropathy and nephropathy were significantly reduced 5 years after transplantation compared to the case before transplantation, as assessed using Fisher’s exact test (Table 6). Furthermore, the number of patients with any chronic complications also decreased significantly (Table 6).

### 3.5. Discussion and Conclusions of Anti-Rejection Strategies and Clinical Outcomes

It should be noted that the Swedish group demonstrated that the porcine islets could survive in the human body with standard immunosuppression for the first time. The Chinese group demonstrated with an advanced immunosuppression protocol that transplanted porcine islet could improve the clinical futures of type 1 diabetes. On the other hand, zoonosis should be carefully monitored the patients who received immunosuppressive drugs.

The New Zealand group initiated encapsulated islet xenotransplantation without immunosuppression and without antibiotics. They continuously improved the islet isolation and capsule protocols and eventually proved the clinical significance of encapsulated islet xenotransplantation. This group introduced the idea of the encapsulation of islets for the avoidance of immunosuppression.

The Mexican group demonstrated islet xenotransplantation under the skin using a device, and Sertoli cells showed efficacy and safety. In fact, islet transplantation under the skin has several advantages, including (1) no blood-mediated inflammatory reaction; (2) no portal vein complications, including bleeding and thrombosis; and (3) being removable. Immunomodulation using Sertoli cells without immunosuppressive drugs can avoid immunosuppression-related side effects. Therefore, porcine islet and Sertoli cell co-transplantation under the skin, using transplant devices, should be a potentially useful approach.

On the other hand, islet xenogeneic transplantation is not effective compared to allogeneic islet transplantation. Because current porcine islet isolation protocol could provide high-quality islets, a search for anti-rejection strategies is important.

## 4. Ethical and Regulatory Considerations Relevant to Xenotransplantation

### 4.1. Ethical Considerations

In general, the Belmont report outlined ethical principles for conducting experimental treatment in humans [39]. Ethical principles include respect for person, which requires informed consent, beneficence, which requires risk and benefit assessment, and justice, which requires equitable selection of research subjects [40]. Xenotransplantation requires specific ethical considerations due to its possible zoonosis, which could be a public risk. The Ethics Committee of the International Xenotransplantation Association outlined ethical considerations when initiating clinical islet xenotransplantation, as follows [41,42].

The need for a favorable risk/benefit assessment based on evidence of strong efficacy data in relevant pre-clinical proof of consent studies in the primate.The patients lack reasonable alternative treatments.The need to minimize the risk of infection by providing the highest biosafety standards for source animals.The need for long-term monitoring of and restrictions on xenograft recipients and, possibly, their close contacts.The welfare aspects related to the use of animalsConfidentiality and privacy issues associated with xenotransplantation. The potential risk of infection related to xenotransplantation goes beyond the patients and requires public authorization.Appropriate authorization and oversight of the xenotransplantation should be conducted at institutional and national levels.The risk of xenotourism, which is the travel of prospective xenotransplant candidates to countries where xenotransplantation procedures are less stringently regulated.Ulysses contracts for xenotransplantation [42]. Because the recipients require life-long monitoring, they need to waive their right to withdraw from the clinical trial.

### 4.2. Regulatory Considerations

Regarding regulatory frameworks, the WHO provided The Changsha Communique, which consists of 10 principles, in 2008.

The summary of the principles is as follows [43].

Successful xenotransplantation has the potential to treat a wide range of serious diseases.Potentially, animals could provide a plentiful supply of cells, tissues, and organs for transplantation. Genetic modification of the animals may improve efficacy. Animals used in xenotransplantation should be from a closed herd breed and housed in a well-controlled, pathogen-free environment.Xenotransplantation is a complex process, and there is a risk of developing serious or novel infections.Because of the wider community risks, xenotransplantation in clinical trials needs to be effectively regulated.Because of the community risk, there should be a high expectation of benefit to balance the risk.Proposers of xenotransplantation clinical trials must be able to clearly justify carrying out a particular trial on a specific patient population. Patient selection should occur on the basis of informed consent.Participation in xenotransplantation will require the long-term storage of animal and patient samples, as well as records.Medical teams must have the appropriate expertise. There must be a system in place for vigilance and surveillance with contingency plans to identify and respond to any indication of xenotransplantation-related infection.There needs to be a global system for exchanging information and preventing unregulated xenotransplantation.The public sector should be encouraged to support xenotransplantation research and development.

A WHO global consultation on regulatory requirements for xenotransplantation clinical trials was held in China in 2018 [44]. The new recommendations were added into the Changsha Communique, and they propose quality control measures and standards for genetically modified pigs and indicate that the devices and biomaterials should comply with current international standards.

Among the requirements, establishing a well-controlled, pathogen-free environment for source animals is challenging [45]. In fact, only New Zealand [19], China [28], and, recently, US groups have successfully provided clinical-grade pigs under the regulation. In other words, only three countries have experiences of providing clinical-grade pigs under the regulation. Therefore, the limited number of donor sources is a current issue in the expansion of xenotransplantation.

## 5. Discussion

Allogenic islet transplantation is considered the standard therapy for unstable type 1 diabetic patients. The aim of allogenic transplantation is currently not to achieve an insulin-free status but to avoid severe hypoglycemia and maintain excellent blood glucose levels [4].

Considering the huge number of type 1 diabetic patients, it is apparent that there is a shortage of human organ donors. To overcome the shortage of donors, clinical islet xenotransplantation has been conducted using fetal or neonatal porcine islets.

The difference in structure between human insulin and porcine insulin is only one amino acid. After the discovery of insulin in 1922, porcine insulin was used to treat type 1 diabetic patients for 60 years. Therefore, porcine insulin is a well-established treatment, which suggests that porcine islet transplantation should also be a viable treatment.

The Swedish group opened the door to this concept based on the idea that islet cell-like clusters (ICC) in fetal rodent tissue and human pancreatic tissue can be utilized to treat diabetes. Then, Korbutt and his colleague established a neonatal porcine islet isolation protocol, which can provide stable, high quantities of quality ICCs. This is critically important for islet transplantation. Since then, ICCs or neonatal porcine islets have been used clinically.

To prevent immunological rejection, the New Zealand team used microencapsulation, while the Mexican group used Sertoli cells. Both groups have shown safety and clinical benefits, including some cases being insulin-free. The Chinese group used Tregs and immunosuppressive drugs, and their patients’ status was substantially improved. Further clinical trials are important to improve and establish islet xenotransplantation.

However, clinical trials of islet xenotransplantation have only been conducted in limited groups. This limitation is mainly caused by the inadequate availability of source animal facilities that can supply clinical-grade pigs. In fact, the New Zealand facility provided clinical-grade pig islets for the New Zealand trial, the Russian trial, the Mexican trial, and the Argentine trial, while the Chinese facility provided clinical-grade pig islets for the Chinese trial. In other words, only two facilities provided clinical-grade pig islets. This limited number of facilities is due to the high cost of building and maintaining such facilities [45], and, possibly, uncertainty regarding the future of xenotransplantation.

Recently, a clinical trial of xenogeneic heart transplantation using a gene-modified pig opened a new door for xenogeneic transplantation [46]. This might encourage the creation of a clinical-grade pig facility.

In order to accelerate xenotransplantation, the ‘Medical Porcine Development Organization’ was established in Japan on December 1st 2023 at the Kobe University School of Medicine. This organization aims to create new medical care protocols and contribute to the health and welfare of the population by establishing and improving the technology to produce pigs that can be used as medical raw materials. Currently, this organization works to connect agriculture teams and medical teams, aiming to initiate islet xenotransplantation in Japan.

Traditionally, germ-free pigs were created through hysterotomy or hysterectomy, followed by rearing in a germ-free environment [47,48]. As long as pregnant pigs are negative for vertically transmitted pathogens, this technology can theoretically create germ-free piglets. The use of neonatal piglets as donors can result in a shorter rearing period in a smaller, germ-free environment, which can lead to cost reductions. Consequently, it might be possible to scale up the production of source animals. Once the technology for the creation of a large quantity of clinical-grade pigs is established, the issue of donor shortages will be resolved. Then, with sufficient quantities of islets, islet transplantation could aim to reach an insulin-free status. Even if the transplanted islets deteriorate, they can be supplemented to restore their function. Previously, we performed an allogeneic islet transplantation 5 years after the first transplantation and achieved an insulin-free status [49]. Therefore, repeatable islet transplantation may be an eventual objective for islet transplantation. The scaling up of clinical-grade pig production will enable us to overcome the donor shortages, which could contribute to the health and welfare of the population.

## 6. Conclusions

Xenogeneic islet transplantation has been conducted, and it has continuously shown safety and efficacy. The accumulation of clinical data is critically important in advancing this field. The limited number of clinical-grade pigs is the main issue at present, which should be resolved by collaborative work between agriculture teams and medical teams.

## Figures and Tables

**Figure 1 cells-13-00828-f001:**
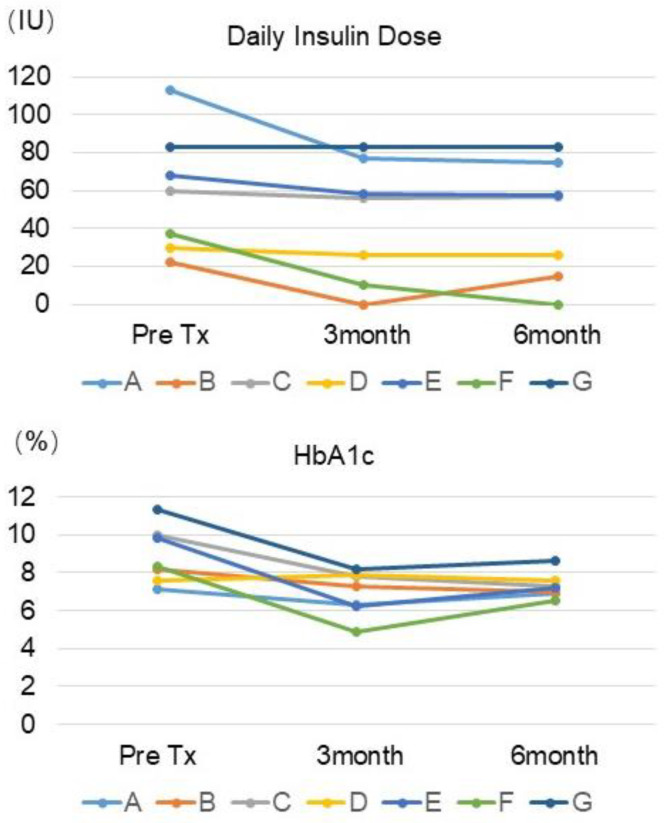
Daily insulin doses of patients who received encapsulated porcine islets (**upper graph**) and HbA1c (**lower graph**) pre-transplantation and 3 months and 6 months after transplantation. Two patients achieved insulin independence, and all patients except for patient D experienced reduced HbA1c after transplantation.

**Table 1 cells-13-00828-t001:** Advantages and disadvantages of allogeneic and xenogeneic islet transplantation.

	Advantages	Disadvantages
Allogeneic islet transplantation	Established treatment for unstable type 1 diabetic patients Majority of countries consider this the standard therapy	Donor shortages Unhealthy brain-dead donors and uncomprehensive screening for pathogens Emergent islet isolation and transplantation Unstable islet isolation Necessity for life-long immunosuppression
Xenogeneic islet transplantation	Possible sufficient donor supply Healthy donors and comprehensive screening for pathogens Routine islet isolation and transplantation Stable islet isolation with fetal or neonatal pancreas Encapsulation technology can eliminate immunosuppression	Limited clinical experience and still experimental Possible zoonosis Xenogeneic islets have strong immunity compared to allogeneic islets Necessity for donor source facilities Need for appropriate ethical and regulatory framework

**Table 2 cells-13-00828-t002:** Islet isolation methods and their clinical application.

Country	Donor	Mince and Digest Method	Culture Method	Yield	Clinical Transplantation	References
Sweden	Human 10–26-week-old fetuses	Minced into fragments of <2 mm^3^ and 2.5 mg/mL collagenase	RPMI 1640 medium + FCS or HS 7 days	148 (per panc with FCS), 980 (per panc with HS)	Not performed	[13]
Sweden	Porcine 51–77-day-old fetuses	Minced into fragments of 1–2 mm^3^ and 12.5 mg/mL collagenase	RPMI 1640 medium + HS 6 or 7 days	>100,000 (per 3 L)	10 type 1 diabetic patients	[14]
New Zealand	Term-gestation piglets	Korsgren protocol	RMPI 1640 + nicotinamide 5 days	19,000 (per gram pancreas)	2 type 1 diabetic patients	[15,16]
Canada	1–3-day neonatal	Minced into fragments of 1–2 mm^3^ and 2.5 mg/mL collagenase	Ham’s F 10 + BSA, L-glutamine + nicotinamide 9 days	Up to 50,000 (per piglet)	Not done	[17]
New Zealand	7–21-day neonatal	Korbutt method, ETK pancreas preservation, Ricordi method	RPMI + HA + nicotinamide Spinner flask 80–90 rpm 3 days, followed by 2–3 weeks of culture	Not shown	8, 16, 21, and 12 type 1 diabetic patients in Russia, New Zealand, Argentina, and Mexico	[18,19,20,21,22,23]
Mexico	5–9-day neonatal	1.5 mg/mL Liberase HI + 150 μ/mL DNase I + 27 μg/mL xylocaine	RPMI 1640 + HS + nicotinamide 4 days	290,730 IEQ per piglet	11 type 1 diabetic patients	[24,25,26,27]
China	Neonatal	Not shown	Not shown	Not shown	10 type 1 diabetic patients	[28]

**Table 3 cells-13-00828-t003:** Anti-immunological rejection strategies, transplant sites, and major findings.

Clinical Trials	Anti-Immunological Rejection Method	Transplant Site	Major Findings	References
Swedish cases	Immunosuppression drugs	Intraportal, kidney capsule	Positive porcine C-peptide for 200–400 days	[8,10]
China trial	Tregs + immunosuppression drugs	Intrahepatic	Substantial improvement in patients’ condition	[28]
New Zealand first trial	Alginate PLL capsule	Abdominal cavity	Reduction of insulin dose and HbA1c with positive porcine C-peptide	[15,16]
Russian trial	Alginate PLO capsule	Abdominal cavity	Achieved insulin-free status in two cases (2/8)	[20]
New Zealand second trial	Alginate PLO capsule	Abdominal cavity	Reduction in number of undetected hypoglycemia events	[21]
Argentine trial	Alginate PLO capsule	Abdominal cavity	Reduction in insulin dose, HbA1c, and undetected hypoglycemia	[23]
Mexican trial	Sertoli cell	Subcutaneous device	Achieved insulin-free status in two patients (2/23); one patient had long-term insulin-free period. All patients showed positive porcine C-peptide	[24,25,26]

**Table 4 cells-13-00828-t004:** Clinical outcomes of New Zealand trial.

	Average Unaware Hypoglycemia per Month		Average Daily Insulin Doses (IU)		Average HbA1c (%)	
	Pre-Transplant	1 Year after Transplant	Pre-Transplant	1 Year after Transplant	Pre-Transplant	1 Year after Transplant
Group 1 (*n* = 4)	11.5	4.8	49.5	37.4	7.38	7.35
Group 2 (*n* = 4)	13.0	2.8	31.1	29.5	7.65	7.63
Group 3 (*n* = 4)	9.8	9.0	41.8	37.9	7.53	7.80
Group 4 (*n* = 2)	24.5	14.0	56.6	59.9	7.30	6.60

**Table 5 cells-13-00828-t005:** Clinical outcome of Argentine trial.

	Average Unaware Hypoglycemia per Month		Average Daily Insulin Doses (IU)		Average HbA1c (%)	
	Pre-Transplant	600 Days after Transplant	Pre-Transplant	600 Days after Transplant	Pre-Transplant	600 Days after Transplant
Group 1 (*n* = 4)	22.3	25.5	58.6	51.0	9.3	7.7
Group 2 (*n* = 4)	43.5	16.4	59.1	43.4	8.4	6.6

**Table 6 cells-13-00828-t006:** Diabetic chronic complications before and 5 years after islet xenotransplantation.

	Before Transplant	Five Years after Transplant	*p* Values
Neuropathy	5/21 (23.8%)	0/21 (0%)	<0.05
Retinopathy	5/21 (23.8%)	1/21 (4.8%)	0.09
Nephropathy	8/21 (38.1%)	2/21 (9.5%)	<0.05
Overall chronic complications	14/21 (66.7%)	2/21 (9.5%)	<0.001

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
