# Peer review of "Clinical Islet Xenotransplantation: Development of Isolation Protocol, Anti-Rejection Strategies, and Clinical Outcomes"

_cells, 2024, doi:10.3390/cells13100828_

Round 1
Reviewer 1 Report (Previous Reviewer 2)
Comments and Suggestions for Authors
Though the authors have tried to improve the manuscript, but here I would like to gain suggest them to go through any recent review of the journal and format the manuscript accordingly.
Mind that this is a review not a research article, no need of a separate introduction and discussion part.
So, please revise the manuscript gain.
Author Response
Though the authors have tried to improve the manuscript, but here I would like to gain suggest them to go through any recent review of the journal and format the manuscript accordingly.
Mind that this is a review not a research article, no need of a separate introduction and discussion part.
So, please revise the manuscript gain.
Authors reply: Thank you for your advice for improving the manuscript.
In the introduction, we describe the structure of this manuscript, namely, isolation method, anti-rejection strategy and clinical outcomes, and ethical and regulatory consideration.
Based on this structure, we re-organized entire manuscript.
We removed separate introduction and discussion as we agree the reviewer’s comment.
The first topic is the establishment of islet isolation protocol, the second topic is anti-rejection strategy and clinical outcome and the third topic is ethical and regulatory considerations. The ethical consideration is requested by one of the reviewers.
We greatly appreciate the reviewer’s suggestion and we feel more conformable for this style.
We hope this major modification can be acceptable for the reviewer.
Reviewer 2 Report (New Reviewer)
Comments and Suggestions for Authors
The manuscript entitled " Clinical Islet Xenotransplantation—Development of Isolation Protocol, Anti-Rejection Strategies, and Clinical Outcomes" is focused on an interesting topic with numerous implications, both medical or ethical concerns.
The topic covered in the review is very interesting and current. However, it is also a highly debated topic from an ethical point of view, as public opinion is very sensitive to the use of animals for medical purposes. The authors do not address this last aspect, to which at least one paragraph should be dedicated, addressing the topic and describing the possible scenarios related to bioethical concerns.
In general, the work needs to be reviewed from different points of view.
In the current format the paper cannot be published.
The authors provide a lot of information and, in some cases, the discussion is not very clear and linear.
There are numerous critical issues in this review, especially regarding the structure and presentation of the data.
First, the authors should better describe what kind of approach they used to write the work. The paper presents different types of studies without describing the common thread that links them. The reader gets lost and it is not clear what the authors' target is.
Introductions and discussions that the authors have included for the different covered topics are lacking in detail and unclear. There are references to paper/protocols that are not well organized and described.
Since the authors discuss different protocols and approaches, they should indicate this working methodology in the general introduction, then, in each introduction they should describe the procedures presented and what the purpose is.
Line 54 Among the disadvantages, the ethical aspect linked to the use of animals as organ donors should also be mentioned.
Line 66 This is an introductory paragraph to the topics covered in the paragraphs below. Therefore, the methods that are intended to be discussed should be indicated in the text.
Line 94 This sentence is taken out of context. The results obtained with the two different supplements should be indicated and the possible mechanism of action of what is indicated in the sentence should be explained.
Line 151 Results are missing.
Line 170 It makes no sense to insert a paragraph for a protocol about which you have no information. It can also just be mentioned in the introduction.
Line 192 The introduction is too short and does not provide any general information regarding the topics that will be addressed in the following paragraphs. It is necessary to better detail the topics that will be covered.
Line 435 This statement does not take into consideration all aspects related to the potential zoonotic risk of transplanting animal-derived material which is not comparable to the preparation of an animal-derived drug.
Author Response
The manuscript entitled " Clinical Islet Xenotransplantation—Development of Isolation Protocol, Anti-Rejection Strategies, and Clinical Outcomes" is focused on an interesting topic with numerous implications, both medical or ethical concerns.
The topic covered in the review is very interesting and current. However, it is also a highly debated topic from an ethical point of view, as public opinion is very sensitive to the use of animals for medical purposes. The authors do not address this last aspect, to which at least one paragraph should be dedicated, addressing the topic and describing the possible scenarios related to bioethical concerns.
Authors reply: Thank you very much for this important topic. We put the ethical aspect as one of the disadvantages of xenotransplantation and mentioned in the discussion. Line 448-
In general, the work needs to be reviewed from different points of view.
In the current format the paper cannot be published.
The authors provide a lot of information and, in some cases, the discussion is not very clear and linear.
There are numerous critical issues in this review, especially regarding the structure and presentation of the data.
First, the authors should better describe what kind of approach they used to write the work. The paper presents different types of studies without describing the common thread that links them. The reader gets lost and it is not clear what the authors' target is.
Authors reply: We also realize the target is not clear. Therefore we put the sentences in the introduction Line 60-
One of the unique advantages of xenogeneic islet transplantation is the stable islet isolation method, which shows no failure when using fetal or neonatal pancreata. In fact, the stable islet isolation method was one of the reasons for the performance of the first islet xenotransplantation using fetal pancreata (8). As a matter of fact, even with the advanced human islet isolation method, the success rate of human islet isolation was found to be only 52.5% (170/324) (9). Therefore understanding the development of fetal or neonatal porcine islet isolation method should be important.
On the other hand, overcoming the xenogeneic rejection is challenging. Anti-rejection drugs, encapsulation and Sertoli cells were used for clinical islet xenotransplantation with modest outcomes.
In this review article, at first, we focused on the development of the porcine islet isolation protocol. Thereafter, anti-rejection strategies and clinical outcomes are discussed. Lastly, we mentioned the ethical and regulatory considerations for clinical xenotransplantation.
Introductions and discussions that the authors have included for the different covered topics are lacking in detail and unclear. There are references to paper/protocols that are not well organized and described.
Since the authors discuss different protocols and approaches, they should indicate this working methodology in the general introduction, then, in each introduction they should describe the procedures presented and what the purpose is.
Authors reply: we appreciate your guide. We reorganize in order to clarify the purpose of the sections.
Line 54 Among the disadvantages, the ethical aspect linked to the use of animals as organ donors should also be mentioned.
Authors reply: We agree and we put ethical and regulatory aspect as section 4.
Line 66 This is an introductory paragraph to the topics covered in the paragraphs below. Therefore, the methods that are intended to be discussed should be indicated in the text.
Authors reply: We agree this is necessary. We put phrases in line 60-
One of the unique advantages of xenogeneic islet transplantation is the stable islet isolation method, which shows no failure when using fetal or neonatal pancreata. In fact, the stable islet isolation method was one of the reasons for the performance of the first islet xenotransplantation using fetal pancreata (8). As a matter of fact, even with the advanced human islet isolation method, the success rate of human islet isolation was found to be only 52.5% (170/324) (9). Therefore understanding the development of fetal or neonatal porcine islet isolation method should be important.
On the other hand, overcoming the xenogeneic rejection is challenging. Anti-rejection drugs, encapsulation and Sertoli cells were used for clinical islet xenotransplantation with modest outcomes.
In this review article, at first, we focused on the development of the porcine islet isolation protocol. Thereafter, anti-rejection strategies and clinical outcomes are discussed.
Line 94 This sentence is taken out of context. The results obtained with the two different supplements should be indicated and the possible mechanism of action of what is indicated in the sentence should be explained.
Authors reply: We apologize we misunderstood this paper. They used only one protocol at this time point. We corrected them. Thank you for pointing out this mistake.
Line 151 Results are missing.
Authors reply: We put results. They assessed viability using AO stain. The viability was 98%, 99% and 99% after 1, 2, and 4 weeks culture respectively. Line 159
Line 170 It makes no sense to insert a paragraph for a protocol about which you have no information. It can also just be mentioned in the introduction.
Author reply: We realized the protocol was already mentioned and this paragraph is not needed. We deleted. .
Line 192 The introduction is too short and does not provide any general information regarding the topics that will be addressed in the following paragraphs. It is necessary to better detail the topics that will be covered.
Authors reply: We realize this is very important. We significantly put more information as follows. Line 189-
The first clinical trial of the Swedish group used ICCs due to the stable islet isolation method. In general, success rate of human islet isolation is approximately 50% (9), adult porcine islet isolation is more difficult compared to human islet isolation (29). Stable islet isolation is critically important for large scale clinical trials or even for industrialization in future. Therefore, understanding the development of fetal or neonatal porcine islet isolation is important.
In order to increase the islet yield, Korbutt and colleagues used a neonatal pancreas instead of a fetal pancreas. This is simply, the size of neonatal pancreas is bigger than the fetal pancreas, and however, this step has significant impact to provide enough amount of islets with less donors. Importantly, the Korbutt islet isolation method becomes a current gold standard for neonatal porcine islet isolation.
The New Zealand group initially used fetal pancreas and then introduced the Korbutt protocol to increase the islet yields and provided neonatal porcine islets for several clinical trials. Since quality and quantity of islet is critically important they continuously improves the neonatal porcine islet isolation method. Further modifications including the use of ETK solution for pancreas preservation, the Ricordi method for pancreas digestion, and nicotinamide for culture are important advancements in neonatal porcine islet isolation.
In conclusion, current neonatal porcine islet isolation method could provide stable high quantity and quality of islets for clinical trials.
Line 435 This statement does not take into consideration all aspects related to the potential zoonotic risk of transplanting animal-derived material which is not comparable to the preparation of an animal-derived drug.
Author reply: We agree we should mention about the zoonotic risk which require secure donor source and regulatory framework. To explain this we added one chapter to describe ethical and regulatory consideration.
Round 2
Reviewer 2 Report (New Reviewer)
Comments and Suggestions for Authors
The authors reviewed the paper as indicated and it can be considered ready for publication.
There are some grammatical errors to correct. A linguistic revision is recommended.
This manuscript is a resubmission of an earlier submission. The following is a list of the peer review reports and author responses from that submission.
Round 1
Reviewer 1 Report
Comments and Suggestions for Authors
The manuscript needs a lot more work - there are a lot of grammatical and typographical errors.
Description of how xenotransplatation works and the clinical implications of it should be discussed in the beginning. For eg: do patients need any antibiotics etc.
It feels like more of a process review of islet extraction and cell culture. We suggest that you re-write it within that scope and resubmit.
English will need to be improved.
Author Response
-
- Comments and Suggestions for Authors
The manuscript needs a lot more work - there are a lot of grammatical and typographical errors.
Authors reply: We apologies this. We submitted our manuscript to English Language Editing Services.
Description of how xenotransplatation works and the clinical implications of it should be discussed in the beginning. For eg: do patients need any antibiotics etc.
Authors reply: We put the explanation of porcine insulin in discussion (Discussion line 429-432). We also added the information about the transplantation at the end of each clinical trials for the explanation of the clinical implications.
It feels like more of a process review of islet extraction and cell culture. We suggest that you re-write it within that scope and resubmit.
Authors reply: To clear the scope, the title was changed to Islet Xenotransplantation –Development of Isolation Protocol, Anti-Rejection Strategies, and Clinical Outcomes- Then we re-write it within that scope.
Comments on the Quality of English Language
English will need to be improved.
Author reply: We apologies this. We submitted our manuscript to English Language Editing Services.
Reviewer 2 Report
Comments and Suggestions for Authors
The review article "Islet Xenotransplantation" is a good attempt to compile the literature on exploring the prospects of Islet cells in regenerative medicine especially for the treatment of type-1 diabetes. The manuscript is well conceptualized and written. However, I have following suggestions/concerns:
1. Manuscript needs to be reorganized into separate sections like isolation of islets, experimental transplantation in different species, clinical applications etc., and have their point of discussion at that point itself instead of having a separate discussion section.
2. Also include the conclusion separately.
3. There are a few grammatical/ organizational suggestions encrypted in the attached edited manuscript, that may be included.

The English language is fine/ no issue; barring a few as indicated in the edited manuscript.
Author Response
- Comments and Suggestions for Authors
The review article "Islet Xenotransplantation" is a good attempt to compile the literature on exploring the prospects of Islet cells in regenerative medicine especially for the treatment of type-1 diabetes. The manuscript is well conceptualized and written. However, I have following suggestions/concerns:
- Manuscript needs to be reorganized into separate sections like isolation of islets, experimental transplantation in different species, clinical applications etc., and have their point of discussion at that point itself instead of having a separate discussion section.
Author reply: We reorganized all section starting from introduction, islet isolation protocol, anti-rejection strategies and clinical outcomes.
We also put discussion at the end of each section.
- Also include the conclusion separately.
Author reply: We put conclusive statements at the end of each section.
- There are a few grammatical/ organizational suggestions encrypted in the attached edited manuscript, that may be included.
Author reply: We apologies this. We submitted our manuscript to English Language Editing Services.
Reviewer 3 Report
Comments and Suggestions for Authors
The manuscript of Matsumoto et al. is aimed at providing a comprehensive review concerning the current knowledge about islet xenotransplantation. Authors divide their manuscript into a first part (Introduction) that provides a general overview about the history of allogeneic islet transplantation acceptance as a therapeutic option for patients with Type 1 diabetes. Authors then divide the manuscript into chapters describing the clinical trials employing xenotransplantation approaches. Authors complement their manuscript with 2 Figures and a 3 Tables summarizing the clinical trials and the results of one trial conducted in Mexico. The division of the manuscript is not completely clear; while chapters 2 and 5 are clearly marked as “Swedish cases” and “Chinese trial” other chapters (3 and 4) are not distinguished by any titles. Authors cite 39 references to provide a complete overview. This reviewer notes the quality work that Authors invested in the manuscript to provide very detailed descriptions of each trial and method discussed.
The theme is of the manuscript is original and current, and of interest for the readers of Cells. The text, however, needs to be revised to provide a more comprehensive division and a logical flow.
This reviewer has the following concerns to address by the Authors before the manuscript could be considered for acceptation.
Main issues
1. The introduction part should provide more information regarding the criteria for acceptation of islet transplantation for Type 1 diabetic patients.
2. The introduction part should contain a Table with the advantages and disadvantages of allogeneic transplantation and how these issues are addressed by xenotransplantation (e.g. islet availability, immunogenicity)
3. The Figures do not provide any meaningful information and should be discarded.
4. There is no Table 3 (Table 4 follows Table 2).
5. Instead of the first Authors’ names, Table1 should have an additional column adding the references of the studies that are described in each line. Similarly, the references should be added in Table 2.
6. Authors mention a novel initiative “Medical Porcine Development Organization” in Japan. It seems to be a very novel and innovative initiative. Authors should provide more information: establishment date, institutions participating in this organization and possible current achievements.
7. The text should be revised for grammatical, linguistic, and typing errors. The following examples do not constitute an exhaustive list of issues but rather highlight the types of concerns that need to be addressed by the Authors.
e.g.
- Page 2, Line 72: “Compared to FCS, human serum could significantly increase ICC yields.” Please provide precise information: percentage of yield improvement and information what reasons Authors of the study in question provided as the reason for increased yield.
- Page 2, Line 62: “grass tubes” correct to “glass tube”.
- Page 3, Line 90: “However, they implanted the ICCs under the kidney capsule of non-diabetic nude mice 90 and after 4 weeks, they found frequent insulin and glucagon-positive cells”. Please precise the number or percentage of cells compared to total islet cell numbers.
- Page 9, Line 311: “They also transplanted autologous Treg 2x106/kg”. This sentence seems to be unfinished.
Comments on the Quality of English Language
English usage needs some revision, mostly grammatical errors and mistypings.
Author Response
- Comments and Suggestions for Authors
The manuscript of Matsumoto et al. is aimed at providing a comprehensive review concerning the current knowledge about islet xenotransplantation. Authors divide their manuscript into a first part (Introduction) that provides a general overview about the history of allogeneic islet transplantation acceptance as a therapeutic option for patients with Type 1 diabetes. Authors then divide the manuscript into chapters describing the clinical trials employing xenotransplantation approaches. Authors complement their manuscript with 2 Figures and a 3 Tables summarizing the clinical trials and the results of one trial conducted in Mexico. The division of the manuscript is not completely clear; while chapters 2 and 5 are clearly marked as “Swedish cases” and “Chinese trial” other chapters (3 and 4) are not distinguished by any titles. Authors cite 39 references to provide a complete overview. This reviewer notes the quality work that Authors invested in the manuscript to provide very detailed descriptions of each trial and method discussed.
The theme is of the manuscript is original and current, and of interest for the readers of Cells. The text, however, needs to be revised to provide a more comprehensive division and a logical flow.
Authors reply: We agree this comment. We re-organized and made a comprehensive division and a logical flow.
This reviewer has the following concerns to address by the Authors before the manuscript could be considered for acceptation.
Main issues
- The introduction part should provide more information regarding the criteria for acceptation of islet transplantation for Type 1 diabetic patients.
Authors reply: We put more information about type 1 diabetic patients in the introduction section (31-40).
- The introduction part should contain a Table with the advantages and disadvantages of allogeneic transplantation and how these issues are addressed by xenotransplantation (e.g. islet availability, immunogenicity)
Author reply: We appreciate this idea and create a new table 1.
- The Figures do not provide any meaningful information and should be discarded.
Author reply: We deleted the figures.
- There is no Table 3 (Table 4 follows Table 2).
Author reply: We apologized this mistakes and corrected.
- Instead of the first Authors’ names, Table1 should have an additional column adding the references of the studies that are described in each line. Similarly, the references should be added in Table 2.
Author reply: We added references column in both tables.
- Authors mention a novel initiative “Medical Porcine Development Organization” in Japan. It seems to be a very novel and innovative initiative. Authors should provide more information: establishment date, institutions participating in this organization and possible current achievements.
Author reply: We put more information about this initiative in line 452-458.
- The text should be revised for grammatical, linguistic, and typing errors. The following examples do not constitute an exhaustive list of issues but rather highlight the types of concerns that need to be addressed by the Authors.
e.g.
- Page 2, Line 72: “Compared to FCS, human serum could significantly increase ICC yields.” Please provide precise information: percentage of yield improvement and information what reasons Authors of the study in question provided as the reason for increased yield.
Author reply: We put ‘Compared to FCS, human serum could suppress the growth of fibroblasts and increase ICC yields about sevenfold.’ (100-101)
To provide actual percentage of yield improvement and reasons (suppress the growth of fibroblasts).
- Page 2, Line 62: “grass tubes” correct to “glass tube”.
Author reply: We apologized and corrected.
- Page 3, Line 90: “However, they implanted the ICCs under the kidney capsule of non-diabetic nude mice 90 and after 4 weeks, they found frequent insulin and glucagon-positive cells”. Please precise the number or percentage of cells compared to total islet cell numbers.
Author reply: Original paper did not described detailed number, therefore, we are not able to describe precise number.
- Page 9, Line 311: “They also transplanted autologous Treg 2x106/kg”. This sentence seems to be unfinished.
Author reply: We changed the sentence as follows ‘They used immunosuppressive drugs and autologous regulatory T cells (Treg) for immunosuppression. The patients received 0.087 mg/kg Tacrolimus twice per day, mycophenolate mofetil 1 g twice per day, and belatacept prior to transplantation, day 5, week 2, 4, 8 and 12 10 mg/kg. The dose of autologous Treg was 2 x 106 cells /kg.’(239-254)
Round 2
Reviewer 1 Report
Comments and Suggestions for Authors
While the authors have changed the title which is a lot more fitting than the previous title, section 3 is severely underdeveloped and slightly comes out of nowhere.
If there is a change in manuscript title, there should be a consequent change in article structure in order for it to be effective.
English language seems like it has been improved.
Author Response
While the authors have changed the title which is a lot more fitting than the previous title, section 3 is severely underdeveloped and slightly comes out of nowhere.
Our reply: We agree we need more information in section 3. In order to have profound information we created one figure (figure 1) and two tables (table 4, table 5).
In addition, we put more information in line 250-257, 331-339, 348-352, 360-363

Reviewer 2 Report
Comments and Suggestions for Authors
The authors have completely failed to revise the manuscript as per suggestions.
Comments on the Quality of English LanguageEnglish has improved significantly.
Author Response
The authors have completely failed to revise the manuscript as per suggestions.
Our reply: We deeply apologized this. We sincerely changed according to your suggestions as follows
The review article "Islet Xenotransplantation" is a good attempt to compile the literature on exploring the prospects of Islet cells in regenerative medicine especially for the treatment of type-1 diabetes. The manuscript is well conceptualized and written. However, I have following suggestions/concerns:
- Manuscript needs to be reorganized into separate sections like isolation of islets, experimental transplantation in different species, clinical applications etc., and have their point of discussion at that point itself instead of having a separate discussion section.
Author reply: We reorganized all section starting from introduction line 30-40, islet isolation protocol in line 70-, anti-rejection strategies and clinical outcomes in line 198-. We did not include experimental transplantation in different species because the paper is focused on clinical islet xenotransplantation. To clarify this we changed the title to `Clinical islet xenotransplantation – Development of isolation protocol, anti-rejection strategies, and clinical outcomes. ` We also put discussion at the end of each section.
- Also include the conclusion separately.
Author reply: We put conclusive statements at the end of each section. For islet isolation protocol we put discussion and conclusion in line 186-295, for immunosuppressive drugs we put discussion and conclusion in line 250-256, for New Zealand group trial we put discussion and conclusion in line 374-379, for Mexican trail we put discussion and conclusion in line 430-438.
- There are a few grammatical/ organizational suggestions encrypted in the attached edited manuscript that may be included.
Author reply: We apologies this. We submitted our manuscript to English Language Editing Services.
